# Anti-Metastatic Effects of Standardized Polysaccharide Fraction from *Diospyros kaki* Leaves via GSK3β/β-Catenin and JNK Inactivation in Human Colon Cancer Cells

**DOI:** 10.3390/polym16091275

**Published:** 2024-05-03

**Authors:** Woo-Seok Lee, Ji-Sun Shin, Seo-Yun Jang, Kyung-Sook Chung, Soo-Dong Kim, Chang-Won Cho, Hee-Do Hong, Young Kyoung Rhee, Kyung-Tae Lee

**Affiliations:** 1Department of Pharmaceutical Biochemistry, College of Pharmacy, Kyung Hee University, Seoul 02447, Republic of Korea; leewoosuck7@hanmail.net (W.-S.L.); jsunvet@naver.com (J.-S.S.); tjdbs2357@naver.com (S.-Y.J.); adella76@hanmail.net (K.-S.C.); 2Department of Life and Nanopharmaceutical Sciences, College of Pharmacy, Kyung Hee University, Seoul 02447, Republic of Korea; 3Department of Orthopaedic Surgery, College of Medicine, Hallym University, Hwaseong-si 18450, Republic of Korea; 4Department of Fundamental Pharmaceutical Science, Graduate School, Kyung Hee University, Seoul 02447, Republic of Korea; 5Department of Urology, College of Medicine, Dong-A University, Busan 49315, Republic of Korea; urotan@dau.ac.kr; 6Research Group of Traditional Food, Korea Food Research Institute, Wanju-gun 55365, Republic of Korea; cwcho@kfri.re.kr (C.-W.C.); honghd@kfri.re.kr (H.-D.H.); ykrhee@kfri.re.kr (Y.K.R.)

**Keywords:** *Diospyros kaki*, polysaccharide, epithelial–mesenchymal transition, matrix metalloproteinases, colon cancer cells

## Abstract

A polysaccharide fraction from *Diospyros kaki* (PLE0) leaves was previously reported to possess immunostimulatory, anti-osteoporotic, and TGF-β1-induced epithelial–mesenchymal transition inhibitory activities. Although a few beneficial effects against colon cancer metastasis have been reported, we aimed to investigate the anti-metastatic activity of PLE0 and its underlying molecular mechanisms in HT-29 and HCT-116 human colon cancer cells. We conducted a wound-healing assay, invasion assay, qRT-PCR analysis, western blot analysis, gelatin zymography, luciferase assay, and small interfering RNA gene silencing in colon cancer cells. PLE0 concentration-dependently inhibited metastasis by suppressing cell migration and invasion. The suppression of N-cadherin and vimentin expression as well as upregulation of E-cadherin through the reduction of p-GSK3β and β-catenin levels resulted in the outcome of this effect. PLE0 also suppressed the expression and enzymatic activity of matrix metalloproteinases (MMP)-2 and MMP-9, while simultaneously increasing the protein and mRNA levels of the tissue inhibitor of metalloproteinases (TIMP-1). Furthermore, signaling data disclosed that PLE0 suppressed the transcriptional activity and phosphorylation of p65 (a subunit of NF-κB), as well as the phosphorylation of c-Jun and c-Fos (subunits of AP-1) pathway. PLE0 markedly suppressed JNK phosphorylation, and JNK knockdown significantly restored PLE0-regulated MMP-2/-9 and TIMP-1 expression. Collectively, our data indicate that PLE0 exerts an anti-metastatic effect in human colon cancer cells by inhibiting epithelial–mesenchymal transition and MMP-2/9 via downregulation of GSK3β/β-catenin and JNK signaling.

## 1. Introduction

Colorectal cancer, often referred to as CRC, ranks as the third most common type of cancer globally, with approximately 152,810 new cases and 53,010 deaths reported in 2024 in the United States [1]. The substantial mortality rate associated with CRC is primarily attributed to metastasis, with liver metastasis being the most common [2]. Current CRC treatments include surgical intervention, chemotherapy, and radiation therapy, which often have undesirable and complex side effects. The growing toxicity as well as chemotherapy resistance poses important challenges, implying the importance of the development of anti-metastatic agents to enhance CRC management.

Metastasis is a complex multistep process, with epithelial–mesenchymal transition (EMT) during its early stages playing a pivotal role in driving the malignant behavior of cancer cells. EMT involves phenotypic changes accompanied by alterations in the expression of specific molecular markers E-cadherin, N-cadherin, and vimentin [3]. The extracellular matrix (ECM) acts as a supportive framework, filling the intercellular spaces between cells and providing protection and support. To facilitate metastasis, cancer cells must invade the ECM through the complex roles of matrix metalloproteinase (MMP)-2/9 and tissue inhibitors of metalloproteinase-1 (TIMP-1) [4]. MMP-2/9 acts as a gelatinase, whereas TIMP-1 inhibits MMP-2/9 [5].

*Diospyros kaki* Thumb, commonly known as persimmon, has a longstanding tradition of use in herbal tea and traditional medicine in East Asia to treat various health conditions such as infectious diseases, chronic cough, chronic ulcers, ischemic stroke, hypertension, and coronary-artery-related diseases. Persimmons contain proanthocyanidins and flavonoids, which exhibit diverse pharmacological effects. Flavonoids extracted from persimmon leaves possess antioxidant, antihypertensive, and anti-allergic properties [6]. In addition, persimmon peel extract suppresses platelet-derived-growth-factor-induced cell migration and invasion and MMP-1 expression in human aortic smooth muscle cells [7]. The ethanol extract of persimmon leaves markedly inhibited hepatocyte-growth-factor-induced EMT and stemness features in hepatocellular carcinoma by preventing downstream signaling of Met [8].

PLE0, a water-soluble polysaccharide fraction obtained from *D. kaki* leaves through pectinase treatment, exhibits a high ratio of arabinose to galactose and contains several uncommon sugars including acetyl groups, apiosyl, and 2-keto-3-deoxy-d-manno-2-octulosonic acid, suggesting the presence of rhamnogalacturonan (RG)-I and RG-II regions in PLE0 [9]. As pectinase treatment potentially hydrolyzes the homogalacturonan regions of pectin, PLE0 is primarily composed of the RG-I and RG-II regions [9,10,11,12,13]. Our research team has previously reported that PLE0 stimulates the immune system by increasing the gene expression of immunomodulators by activating the Toll-like receptor 2 (TLR2)-mediated NF-κB signaling pathway [14]. Additionally, PLE0 plays a role in regulating epithelial–mesenchymal transition (EMT) by suppressing the expression of type II transforming growth factor beta receptor (TβRII) and the phosphorylation of TβRI in A549 lung cancer cells [15]. Additionally, a polysaccharide fraction (PLE-II) derived from *D. kaki* leaves has been shown to limit angiogenesis by modulating the production of vascular endothelial growth factor (VEGF) and matrix metalloproteinase-9 (MMP-9) through MAPKs and NF-κB p65 signaling pathways [16]. Although the anti-metastatic properties of PLE0 in colon cancer cells have not been previously reported, this study intends to investigate the molecular mechanisms that contribute to PLE0’s anti-metastatic efficacy in the HT-29 and HCT-116 CRC cells.

## 2. Materials and Methods

### 2.1. Reagents

The protocol of PLE0 preparation has been described in previous studies [14,15]. According to our previous report [14], PLE0 primarily consists of neutral sugars such as arabinose, fucose, glucuronic acid, mannose, and rhamnose. The PLE0 was dissolved in phosphate-buffered saline (PBS) and used for subsequent studies. The antibodies are specified in Appendix A.

### 2.2. Cell Culture

HT-29 and HCT-116 cells (human colon carcinoma cell lines) were acquired from the Korean Cell Line Bank (KCLB; Seoul, Republic of Korea) and were maintained in RPMI medium (10% heat-inactivated FBS, 100U/mL penicillin) following the cell culture guideline of KCLB.

### 2.3. Cell Viability

A cell viability assessment was carried out using the MTT assay [17]. The 96-well plate was seeded with cells (3 × 10^4^/100 μL/well) and incubated for the assay. Subsequently, 100 μL of PLE0 (0, 10, 25, or 50 μg/mL) was treated on each well, and the cells were incubated for an additional 48 h. Afterward, MTT stock solution (20 μL, 5 mg/mL) was put on each well and reacted at 37 °C for 4 h. The optical density of the formazan blue solution was measured at 540 nm after dissolving it in 200 μL DMSO.

### 2.4. Wound-Healing Assay

Confluent cells (2 × 10^6^ cells/mL) were prepared in 6-well plates, as described previously [17]. An area devoid of cells was established by employing a cell scraper with a 13 mm blade width to mechanically remove the center portion of the culture vessel. Treatment with or without PLE0 (0, 10, 25, or 50 μg/mL) for 48 h was subsequently conducted after cleaning the cells with the medium.

### 2.5. Invasion Assay

In a transwell chamber (Corning; Steuben, NY, USA) that was pre-coated with matrigel (Thermo Fisher Scientific, Waltham, MA, USA), cell suspensions were added to the culture medium [17]. Downwells were exposed to PLE0 (0, 10, 25, or 50 μg/mL) and then incubated for 48 h. Afterward, the cells were treated with a Diff Quik kit manufactured by Sysmex Corporation (Kobe, Japan).

### 2.6. qRT-PCR Assay

The Easy Blue^®^ kits from Intron Biotechnology (Intron Biotechnology Inc.; Seoul, Republic of Korea) were used successfully to extract high-quality total cellular RNA. The quantity of RNA was determined using the NanoDropTM 2000/2000c Spectrophotometer from Thermo Fisher Scientific (Waltham, MA, USA). A QuantStudioTM 1 Real-Time PCR Instrument (Thermo Fisher Scientific; Waltham, MA, USA) was utilized to reverse-transcribe and amplify quantified RNA, referencing the previous report [18]. Bioneer (Seoul, Republic of Korea) designed the primer sequences that are detailed in Appendix A.

### 2.7. Western Blotting

As described in our previous report [19], the lysis of cells was accomplished using PRO-PREP protein extraction solution from Intron Biotechnology (Seoul, Republic of Korea) at 4 °C for 20 min. The Bio-Rad protein assay reagent was used to determine the protein concentration. Thirty micrograms of protein was resolved through 8–12% SDS-PAGE gel electrophoresis and subsequently transferred to a PVDF membrane through electroblotting. Densitometric analysis was conducted using the Bio-rad Quantity One^®^ Software, version 4.6.3 (BioRad; Hercules, CA, USA).

### 2.8. Gelatin Zymography Assay

A zymography protease assay was used to measure the activity of MMP-2 and MMP-9, following a previously described method [20]. Following the plating of 2 × 10^6^ cells/mL in 6-well plates for 24 h, aliquots of the media were analyzed through 0.1% gelatin-8% SDS-PAGE electrophoresis. After performing electrophoresis, the gels were cleaned with a solution containing 2.5% Triton X-100 and placed in a reaction buffer consisting of 40 mM Tris-HCl (pH 8.0), 10 mM CaCl_2_, and 0.01% NaN3 at 37 °C for 24 h. Following this, the gel was dyed with Coomassie Brilliant Blue R-250 obtained from Bio-Rand (Hercules, CA, USA).

### 2.9. Transient Transfection and Luciferase Assay

pNF-κB-Luc or pAP-1-Luc reporter vector along with the phRL-TK vector (Promega; Madison, WI, USA) using Lipofectamine LTX™ (Thermo Fisher Scientific; Waltham, MA, USA) was transfected into HT-29 and HCT-116 cells for 18 h. The luciferase activity of each cell was estimated using the luciferase assay (Promega; Madison, WI, USA).

### 2.10. Small-Interfering RNA (siRNA) Transfection

siRNAs targeting human JNK1 and JNK2 were obtained from Thermo Fisher Scientific (Waltham, MA, USA). siRNA transfection was conducted by using an Amaxa cell line nucleofector kit (Lonza; Basel, Switzerland). JNK1 siRNA (100 pM) and JNK2 (100 pM) or control siRNA (100 pM) were added to cells that had been incubated for 24 h and pretreated with or without PLE0.

### 2.11. Statistical Analysis

The mean values obtained from three separate experiments are presented with their standard deviations (SD). Statistical significance was determined via a one-way analysis of variance (ANOVA), followed by a Dunnett’s post hoc test. *p*-values less than 0.05 were determined to be statistically significant.

## 3. Results

### 3.1. PLE0 Inhibits the Migration and Invasion of HT-29 and HCT-116

Testing of the cell viability of PLE0 was conducted on HT-29 and HCT-116 cells using an MTT assay. There were no obvious cytotoxic effects after treatment with PLE0 (10, 25, or 50 μg/mL) in both HT-29 and HCT-116 cells (Figure 1A,B). Next, we conducted wound-healing and invasion assays to investigate whether PLE0 had an inhibitory effect on EMT features. PLE0 dose-dependently suppressed cell migration in HT-29 and HCT-116 cells, estimated as 63.80% and 64.34% inhibition, respectively (Figure 1C). The anti-metastatic activity of PLE0 was further investigated in Matrigel-coated transwells. PLE0 treatment for 24 h resulted in a significant decrease in the number of invasive cells when contrasted with control cells (Figure 2). These data demonstrate that PLE0 reduces the metastatic ability of HT-29 and HCT-116 cells without compromising cell viability. Based on these data, we investigated the mechanism of action of PLE0 in HT-29 cells.

### 3.2. PLE0 Inhibits Mesenchymal and Epithelial Phenotypes in HT-29 Cells

mRNA expression levels of E-cadherin, N-cadherin, and vimentin were determined using qRT-PCR to investigate the effects of PLE0 on EMT markers. As shown in Figure 3A, PLE0 dose-dependently enhanced the E-cadherin mRNA levels and significantly suppressed those of N-cadherin and vimentin. EMT subsequently activated the β-catenin pathway, which aided in metastasis progression [21]. Β-Catenin is usually phosphorylated by active GSK3β and subsequently targeted for proteasomal degradation under normal conditions [22]. Western blot analysis of PLE0-treated HT-29 cells revealed a concentration-dependent reduction in phosphorylated GSK3β (inactive form) and β-catenin levels (Figure 3B). This finding implies that PLE0 hinders the migration and invasion of HT-29 cells by restoring the EMT.

### 3.3. The Activity and Expression of MMP-2 and MMP-9 by PLE0 in HT-29 Cells

The impact of PLE0 on the activity and expression levels of MMP-2 and MMP-9, two major MMPs that mediate ECM degradation, were investigated as the activity of MMP is critical in cancer metastasis [23]. Gelatin zymography revealed that PLE0 significantly suppressed the gelatinolytic activities of MMP-2 and MMP-9 (Figure 4A). To determine whether these reduced activities were attributable to the suppression of their expression, we measured the protein and mRNA levels of MMP-2 and MMP-9 using western blotting and qRT-PCR, respectively. The expression of MMP-2 and MMP-9 was conspicuously reduced at both the mRNA and protein levels by PLE0 (Figure 4B,C). The expression of TIMP-1 was further investigated as TIMP-1 can suppress MMP-2 and MMP-9 activities [24]. PLE0 potently increased the protein and mRNA expression of TIMP-1 (Figure 4D,E). These findings suggest that PLE0 inhibits the activities of MMP-2 and MMP-9, as implied by downregulated gene expression with increased TIMP-1 expression.

### 3.4. PLE0 Suppresses the Activation of NF-κB and AP-1 in HT-29 Cells

To understand the mechanism underlying the suppression of MMPs via PLE0, we examined whether PLE0 alters the transcriptional activities of NF-κB and AP-1. Transient transfection of pNF-κB-luc or pAP-1-luc reporter genes into cells was performed, followed by treatment with PLE0 for 24 h. PLE0 concentration-dependently decreased the promoter activities of both NF-κB and AP-1 (Figure 5A,C). As phosphorylated NF-κB and AP-1 activate downstream target genes at the transcriptional level [25,26], we further investigated the phosphorylation of p65, c-Fos, and c-Jun to reveal the molecular mechanisms of PLE0 action. Consistent with the marked inhibition of the transcriptional activity of NF-κB and AP-1, PLE0 downregulated the levels of p-p65, p-c-Fos, and p-c-Jun (Figure 5B,D).

### 3.5. PLE0 Inhibits Metastasis via JNK Inactivation in HT-29 Cells

Mitogen-activated protein kinases (MAPKs), which include p38, JNK, and ERK1/2, have been linked to the metastatic potential of human colon cancer cells by regulating the activation of various transcription factors and subsequently increasing the expression of metastasis-related genes [27]. Thus, we examined whether PLE0 can regulate MAPK phosphorylation in HT-29 cells using western blotting. PLE0 markedly reduced the expression levels of p-JNK but had no effect on p-p38, p-ERK, or non-phosphorylated p38, JNK, and ERK1/2 levels (Figure 6A). JNK activation is involved in cancer cell migration and metastasis [28]. Wound closure and invasion assays after treatment with SP600125, a specific JNK inhibitor, revealed significantly suppressed cell migration and invasion of HT-29 cells (Figure 6B). Furthermore, endogenous JNK was deleted using RNA interference to investigate the anti-metastatic activity of PLE0 in HT-29 cells (Figure 6C). The silencing of JNK restored the PLE0-suppressed mRNA expressions of MMP-2/9- and PLE0-induced mRNA expressions of TIMP-1 (Figure 6D–F). These results demonstrate that PLE0 inhibits metastasis through the JNK pathway in HT-29 cells.

## 4. Discussion

Cancer remains a critical global health concern despite improved survival rates. However, the complex physiological mechanisms involved continue to limit cancer treatments. Approximately 90% of cancer-related deaths are associated with cancer metastasis, implying an essential role of metastasis in cancer progression, including CRC [29]. As Western dietary habits spread to Eastern countries, the incidence of CRC increased, highlighting the need for safe and effective preventive therapies [30]. Polysaccharides with diverse pharmacological actions and good safety profiles have gained recognition in clinical and functional food applications [31]. This study investigated the anti-metastatic effects of PLE0 in HT-29 and HCT-116 CRC cells to reveal the mechanisms underlying CRC metastasis.

Cancer cells undergo EMT, which involves phenotypic changes characterized by altered expression of E-cadherin, N-cadherin, and vimentin during the early stages of metastasis [32]. Previous studies reported that polysaccharides from Scutellaria barbata inhibit EMT in CRC cells via the Smad2/3 signaling pathway. Notably, these polysaccharides from *S. barbata* had arabinose/galactose ratios similar to those of the RG-1 components [33]. We investigated the effects of PLE0 on EMT-related biomarkers in CRC cells. Specifically, our observations revealed a notable upregulation of E-cadherin expression, along with a concomitant decrease in N-cadherin and vimentin expression. These findings suggest that RG-1 and RG-2, which are considered the main constituents of PLE0, contribute to the inhibition of EMT.

Changes in EMT-related biomarkers induced by PLE0 are closely related to intracellular signaling pathways such as GSK3β and β-catenin, which crucially regulate the transition of epithelial cells with strong mobility into a mesenchymal phenotype during the early stages of metastasis. GSK3β, a multifunctional serine/threonine kinase, plays an important role in numerous cellular processes, including proliferation, differentiation, and survival. GSK3β plays a crucial role in the EMT by regulating the Wnt signaling pathway via β-catenin stability. In the absence of Wnt ligands, GSK3β phosphorylates β-catenin, targeting it for ubiquitination and subsequent proteasomal degradation, preventing cytoplasmic accumulation of β-catenin and its nuclear translocation. Nuclear β-catenin promotes gene expression related to the mesenchymal phenotype and suppresses the characteristic features of epithelial cells, such as E-cadherin [34]. Previous studies reported that the ethanol extract of persimmon leaves markedly inhibits hepatocyte-growth-factor-induced EMT and stemness features in hepatocellular carcinoma by preventing downstream signaling of Met/JNK/c-Jun [8]. Additionally, EMT controlled via PLE0 suppressed the levels of TβRII and phosphorylated TβRI in A549 lung cancer cells [15]. Our results suggest that PLE0 reduces phosphorylated GSK3β levels and enhances β-catenin degradation, which results in increased E-cadherin and reduced N-cadherin and vimentin levels. In line with the previous studies, these findings advocate for the essential roles of GSK3β and β-catenin in EMT [34].

Inhibition of MMP expression and activity is considered a preventive strategy against cancer metastasis. Previous studies reported that polysaccharides derived from Rhizopus nigricans, Lycium barbarum, and Inonotus obliquus have been found to inhibit MMP-2 and MMP-9 activities. Similarly, PLE0 downregulated the expression and activity of MMP-2 and MMP-9, suggesting a mechanism for inhibiting matrix degradation. Furthermore, TIMP-1 is an inhibitor of MMPs, including MMP-2 and MMP-9, and reduces their expression and activity [22]. Our findings also suggest that PLE0 enhances TIMP-1 expression and indicates potential mechanisms for the inhibitory effects on MMP-2 and MMP-9 activities. As considerable evidence has implicated the upregulation of MMP-1, -2, -3, -7, -9, and -13 in the development of human colorectal cancers [9], further studies are needed to investigate the effects of PLE0 on various MMP activities using ITC or thermal shift assays.

Intracellular signaling pathways involving MAPK, NF-κB, and AP-1 are associated with the regulation of MMP-2 and MMP-9 expression [28,35]. Li et al. suggested that cholic acid triggers the expression of MMP-9 in colon cancer cells by activating the MAPK pathway. [28]. Jemaa et al. demonstrated that reversine targets JNK1 in colon cancer cells, leading to the inhibition of migration [36]. In addition, utidelone suppressed the growth of colon cancer cells by modulating ROS/JNK signaling [37]. Similarly, our study revealed that PLE0 led to a decrease in JNK phosphorylation and the suppression of AP-1 and NF-κB transcriptional activities, resulting in decreased MMP-2/-9 expression.

The JNK signaling pathway, a member of the MAPK family, is crucial for cellular processes and has been studied in various cancers, such as breast cancer and CRC. When triggered by stress signals, JNK helps cancer cells spread by regulating cell movement and invasion into tissues. It also plays a role in EMT, making cancer cells more invasive. In addition, JNK affects the activity of enzymes (MMPs) that help cancer cells invade tissues by breaking down the ECM [38]. In this study, a JNK inhibitor prevented cell migration and metastasis. Consequently, it was necessary to examine the role of JNK in the anti-metastatic activity of PLE0 using JNK siRNA. Upon JNK silencing, recovery of the mRNA expression of ECM degradation-related proteins regulated via PLE0 was observed. Based on these results, PLE0 exerted anti-metastatic activity by disrupting JNK signaling. However, we did not clarify the role of JNK signaling in the regulation of EMT-related proteins via PLE0, which is a limitation of this study.

## 5. Conclusions

The findings of this study suggest that PLE0 exhibits a promising anti-metastatic capacity by targeting JNK as a key factor associated with invasion, migration, matrix degradation, and EMT by inhibiting MMP-2/9 in CRC cells. Therefore, these findings have important implications for natural alternatives to prevent and improve colorectal cancer metastasis in clinical cancer therapy. However, further research, including animal experiments, is essential to determine the anti-metastatic action and precise mechanisms of PLE0 and to evaluate its potential for clinical applications in humans.

## Figures and Tables

**Figure 1 polymers-16-01275-f001:**
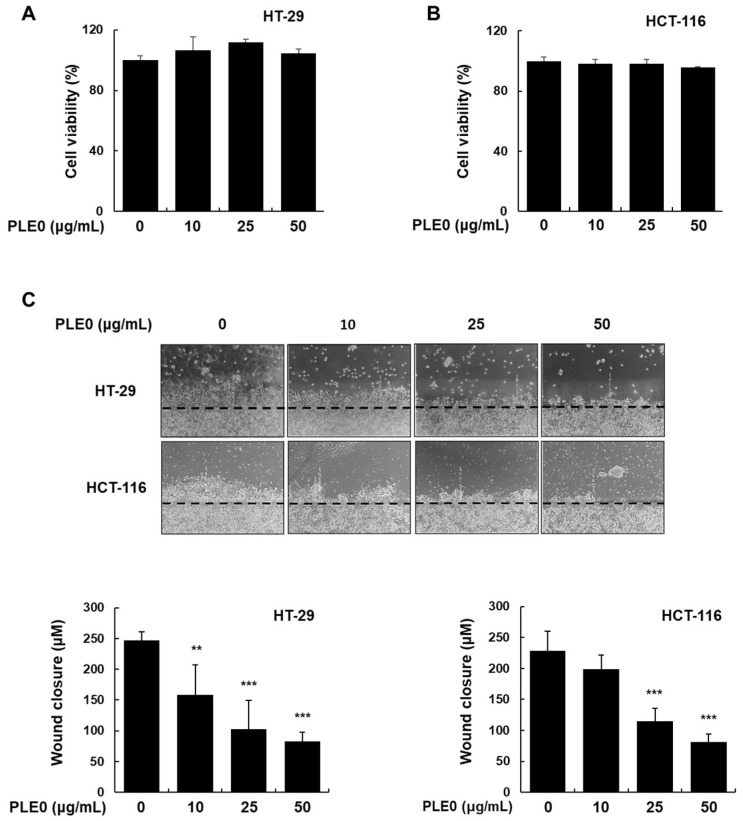
Effects of PLE0 on the migration of HT-29 and HCT-116 cells. (**A**,**B**) Cells were subjected to treatment with the PLE0 at the indicated concentration for 48 h, and cytotoxicity was evaluated using an MTT assay. (**C**) The migratory properties of HT-29 and HCT-116 cells were analyzed using the wound-healing assay. The means ± SD of three separate experiments were reported. The differences between groups were assessed using analysis of variance and Dunnett’s post hoc test to determine significance. ** *p* < 0.01 and *** *p* < 0.001 vs. control.

**Figure 2 polymers-16-01275-f002:**
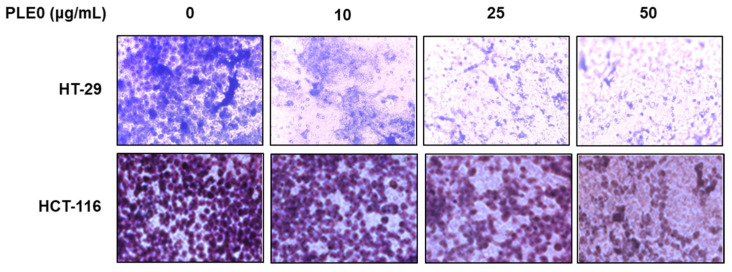
Effects of PLE0 on the invasion of HT-29 and HCT-116 cells. PLE0 was treated at the indicated concentration for 48 h, and the invasive properties of HT-29 and HCT-116 cells were analyzed using invasion assays.

**Figure 3 polymers-16-01275-f003:**
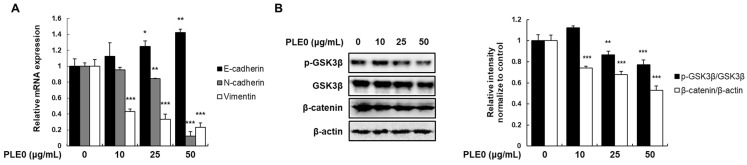
Changes in the expression of the epithelial–mesenchymal transition (EMT) marker and GSK3β/β-catenin signaling in HT-29 cells with PLE0 treatment. (**A**) Treatment with PLE0 at the indicated concentration in the cells for 24 h; E-cadherin, N-cadherin, and vimentin mRNA expressions were then assessed using qRT-PCR. (**B**) Total cellular proteins were harvested and adjusted to detect GSK3β, p-GSK3β, and β-catenin using specific antibodies. The means ± SD of three separate experiments were reported. The differences between groups were assessed using analysis of variance and Dunnett’s post hoc test to determine significance. * *p* < 0.05, ** *p* < 0.01, and *** *p* < 0.001 vs. control.

**Figure 4 polymers-16-01275-f004:**
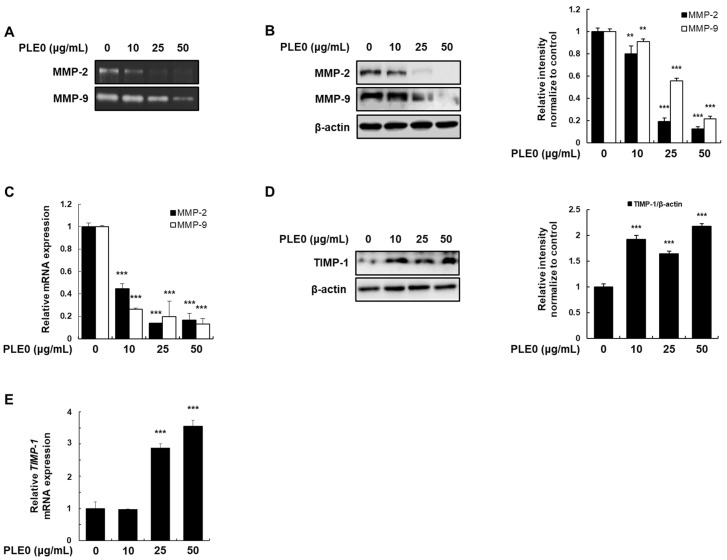
Effects of PLE0 on MMP-2 and MMP-9 proteolytic activity and expression and TIMP expression in HT-29 cells. (**A**) Cells were treated with PLE0 for 24 h and the activity of MMP-2 and MMP-9 was determined using gelatin. (**B**,**D**) Total cellular proteins were harvested and adjusted to detect MMP-2, MMP-9, and TIMP-1 using antibodies. (**C**,**E**) The levels of mRNA for MMP-2, MMP-9, and TIMP-1 were analyzed using gene-specific primers (Appendix A). The mRNA levels of MMP-2, MMP-9, and TIMP-1 were adjusted to β-actin expression. The means ± SD of three separate experiments were reported. The differences between groups were assessed using analysis of variance and Dunnett’s post hoc test to determine significance. ** *p* < 0.01 and *** *p* < 0.001 vs. control.

**Figure 5 polymers-16-01275-f005:**
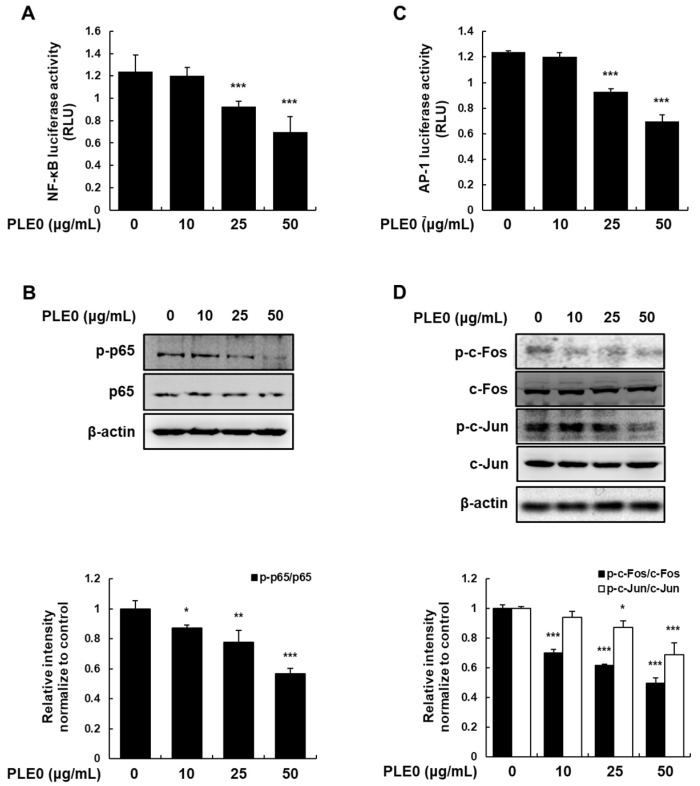
Effect of PLE0 on the activation of NF-κB and AP-1 in HT-29 cells. (**A**,**C**) The NF-κB-Luc and AP-1-Luc reporter vectors were transiently transfected into cells, and an internal control was established using the phRL-TK vector. After applying a prespecified concentration of PLE0 for 24 h treatment, the cells were collected. The assessment of luciferase activity was evaluated utilizing the luciferase assay kit provided by Promega. (**B**,**D**) Total cellular proteins were detected using western blot analysis with specific antibodies. The means ± SD of three separate experiments were reported. The differences between groups were assessed using analysis of variance and Dunnett’s post hoc test to determine significance. * *p* < 0.05, ** *p* < 0.01, and *** *p* < 0.001 vs. control.

**Figure 6 polymers-16-01275-f006:**
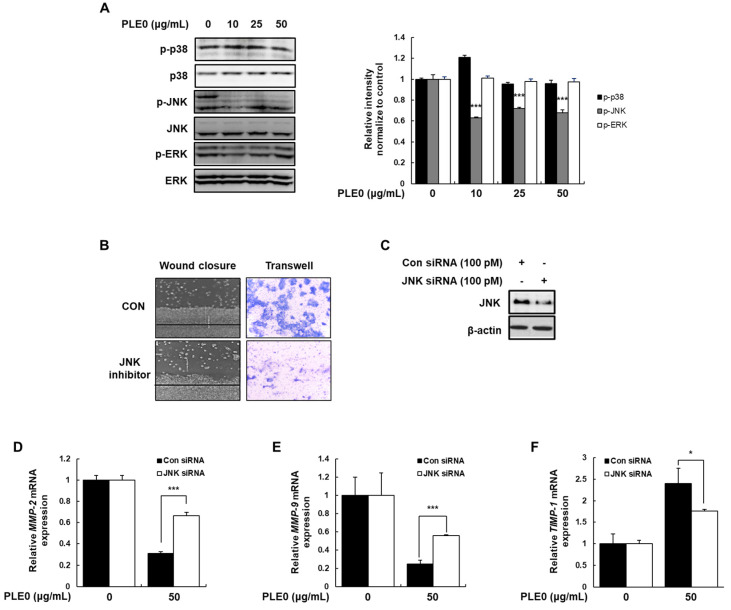
Effects of PLE0 on the JNK signaling pathway in HT-29 cells. (**A**) Cells were treated with PLE0 for 24 h. Western blot analysis was performed on total cellular proteins using specific antibodies. As an internal control, β-Actin was used. (**B**) Cells were treated with SP600125, a specific JNK inhibitor, for 48 h. The migration and invasive traits of HT-29 cells were evaluated through wound-healing and invasion assays, respectively. (**C**) Cells were transfected with negative control and JNK siRNA. Specific antibodies were used to detect JNK and β-catenin in the total cellular proteins that were collected and adjusted. (**D**–**F**) Cells were transfected with 100 pM of negative control and JNK siRNA and then treated with the indicated concentration of PLE0 for 24 h. The mRNA levels were analyzed using gene-specific primers as described in the experimental Section 2.6. The means ± SD of three separate experiments were reported. The differences between groups were assessed using analysis of variance and Dunnett’s post hoc test to determine significance. * *p* < 0.05 and *** *p* < 0.001 vs. control.

## Data Availability

All relevant data are present in the manuscript, figures, and tables.

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
