# Peer review of "Anti-Metastatic Effects of Standardized Polysaccharide Fraction from Diospyros kaki Leaves via GSK3β/β-Catenin and JNK Inactivation in Human Colon Cancer Cells"

_polymers, 2024, doi:10.3390/polym16091275_

Round 1

Reviewer 1 Report

Comments and Suggestions for Authors

Lee and co-workers report that “Anti-metastatic effects of standardized polysaccharide fraction from Diospyros kaki leaves via GSK3β/β-catenin and JNK inactivation in human colon cancer cells.” In their study, they found that PLE0 plays a potential role in targeting JNK as a key factor associated with invasion, migration, matrix degradation, and EMT in CRC cells. I recommend the following minor corrections and suggestions to be made before considering it for publication in the Polymers Journal.

1.      Authors should also perform an ITC or thermal shift assay to examine direct binding with MMP.

2.     Authors should also check the selectivity over other isoforms of MMP.

3.     There are also some typographical mistakes. It should be corrected.

Comments on the Quality of English Language

Minor editing of English language required

Author Response

REVIEWERS' COMMENTS:  

 Reviewer #1: Lee and co-workers report that “Anti-metastatic effects of standardized polysaccharide fraction from Diospyros kaki leaves via GSK3β/β-catenin and JNK inactivation in human colon cancer cells.” In their study, they found that PLE0 plays a potential role in targeting JNK as a key factor associated with invasion, migration, matrix degradation, and EMT in CRC cells. I recommend the following minor corrections and suggestions to be made before considering it for publication in the Polymers Journal.

1 & 2. Authors should also perform an ITC or thermal shift assay to examine direct binding with MMP. Authors should also check the selectivity over other isoforms of MMP

Response to 1 and 2:

Thank you for your valuable comments on our study. In human colorectal cancers, considerable evidence has implicated the overexpression of MMP-1, -2, -3, -7, -9, and -13. Among the MMPs, colon cancers were more often positive by immunohistochemistry for MMP-2 and MMP-9 [1], therefore we have focused on the anti-metastatic effect of PLE0 on the suppression of the proteolytic activity and expression of MMP-2 and MMP-9 and upregulation of the expression of TIMP-1 in HT-29 human colon cancer cells. According to your advice, we have examined the effect of PLE0 on the other MMPs (MMP-1, -3, -7, and -13) and observed that PLE0 only significantly reduced the mRNA expression of MMP-7 (Supplementary Fig.1). Based on the data of this study, in order to obtain more precise research results, we will prepare for further study by expanding the research contents to investigate the effects of PLE0 on various MMPs through ITC or thermal shift assays with pilot scale production of PLE0 prepared by the Korea Food Research Institute to develop CRC therapeutic candidate agents.

3. There are also some typographical mistakes. It should be corrected

Response: To follow your comments, we consulted a professional language editing service (http:// www.editage.co.kr/ Tel: 82-2-1544-9241) and a professional editor revised the whole manuscript carefully (MS no. WOACC_45). Nevertheless, we scrutinized our manuscript and corrected typographical mistakes.

Thank you very much for your advice.

Sincerely yours,

Kyung-Tae Lee, Ph.D.

Department of Pharmaceutical Biochemistry, College of Pharmacy

Kyung-Hee University

Dongdaemun-Ku, Hoegi-Dong 130-701, Seoul, Republic of Korea

Tel: +82-2-9610860, FAX: +82-2-9620860

Reference

  1. Zucker, S.; Vacirca, J. Role of matrix metalloproteinases (MMPs) in colorectal cancer. Cancer Metastasis Rev 2004, 23, 101-117, doi:10.1023/a:1025867130437.

Reviewer 2 Report

Comments and Suggestions for Authors

1. Necessary characterization on the polysaccharides is needed, which is very crucial for the study of bioactivities as well as repeatability.

2. Please provide references for RNA extraction and Western blotting.

3. A control (clinic drug or natural product) is suggested for the bioactivity assay.

4. What 's the solvent for PLE0?

5. The molecular mechanisms of PLE0 need to be compared with current studies.

6. Physical animal models and observation of tumors are recommended for verification.

 7. The section of conclusions should be strengthened, and the innovation and features of the study should be highlighted. 

Author Response

REVIEWERS' COMMENTS:  

 Reviewer #2:

1) Necessary characterization on the polysaccharides is needed, which is very crucial for the study of bioactivities as well as repeatability

Response: To follow your comment, we added the characterization of the PLE0 in 2.1. Reagents of 2. Materials and Methods in the revised manuscript.

2) Please provide references for RNA extraction and Western blotting
Response: To follow your comment, we added the reference for RNA extraction and Western blotting in 2. Materials and Methods of the revised manuscript.

3) A control (clinic drug or natural product) is suggested for the bioactivity assay

Response: I agree with your advice. Now, we are preparing a further study to explore in vivo efficacy and toxicological studies with pilot scale PLE0 production prepared by the Korea Food Research Institute to develop CRC therapeutic candidate agent. In the further study, we will use 2-stearoxyphenethyl phosphocholine [1] as the positive control of bioactivity assay for metastasis progression assessment in CRC cells.

4) What 's the solvent for PLE0?

Response: We added solvent for PLE0 in 2.1. Reagents of 2. Materials and Methods.

5) The molecular mechanisms of PLE0 need to be compared with current studies

Response: Thank you for your comment. Our previous study reported that a pectic polysaccharide fraction isolated from D. kaki leaves inhibits cancer cell proliferation, migration, invasion, and metastasis through the regulation of PI3K/AKT, p38 MAPK, JNK, and NF-κB pathways in HUVECs [2]. PLE0 also regulated EMT by inhibiting the expression of type II TGFβ receptor (TβRII) and phosphorylation of TβRI in A549 lung cancer cells [3]. In addition, PLE0 has demonstrated immunostimulatory effects by activating TLR2/GATA3 signaling in immune cells, thereby contributing potentially impacting cancer progression by immune responses [4]. Although our studies suggest anti-metastasis activities through various mechanisms, the anti-metastatic activity of PLE0 in colon cancer cells has been reported through the downregulation of GSK3β/β-catenin and JNK signaling pathways. We have previously mentioned the various mechanisms of action of PLE0, the molecular mechanisms of PLE0 elucidated in this research extend existing knowledge regarding the anti-metastatic effects of polysaccharides derived from D. kaki leaves. The exact reason why PLE0 exhibits anti-metastasis activity through these various signaling pathways is not known, but it is expected to be a different cancer cell type. This exact mechanism needs to be accurately demonstrated through further in vivo animal investigation.

6) Physical animal models and observation of tumors are recommended for verification.
Response: Thank you for your valuable advice about our study. Although we did only a cell-based study, we will further plan to use an orthotopic model to closely monitor and reliably evaluate primary tumor development, metastatic activity, and therapy response.  

7) The section of conclusions should be strengthened, and the innovation and features of the study should be highlighted.

Response: To follow your comments, we strengthened the novelty and features of this research in the conclusion of the revised manuscript.

Thank you very much for your advice.

Sincerely yours,

Kyung-Tae Lee, Ph.D.

Department of Pharmaceutical Biochemistry, College of Pharmacy

Kyung-Hee University

Dongdaemun-Ku, Hoegi-Dong 130-701, Seoul, Republic of Korea

Tel: +82-2-9610860, FAX: +82-2-9620860

Reference

  1. Park, S.E.; Chung, K.S.; Heo, S.W.; Kim, S.Y.; Lee, J.H.; Hassan, A.H.E.; Lee, Y.S.; Lee, J.Y.; Lee, K.T. Therapeutic role of 2-stearoxyphenethyl phosphocholine targeting microtubule dynamics and Wnt/beta-catenin/EMT signaling in human colorectal cancer cells. Life Sci 2023, 334, 122227, doi:10.1016/j.lfs.2023.122227.
  2. Park, J.Y.; Shin, M.S. Inhibitory Effects of Pectic Polysaccharide Isolated from Diospyros kaki Leaves on Tumor Cell Angiogenesis via VEGF and MMP-9 Regulation. Polymers (Basel) 2020, 13, doi:10.3390/polym13010064.
  3. Lim, W.C.; Choi, J.W.; Song, N.E.; Cho, C.W.; Rhee, Y.K.; Hong, H.D. Polysaccharide isolated from persimmon leaves (Diospyros kaki Thunb.) suppresses TGF-beta1-induced epithelial-to-mesenchymal transition in A549 cells. Int J Biol Macromol 2020, 164, 3835-3845, doi:10.1016/j.ijbiomac.2020.08.155.
  4. Lee, S.G.; Jung, J.Y.; Shin, J.S.; Shin, K.S.; Cho, C.W.; Rhee, Y.K.; Hong, H.D.; Lee, K.T. Immunostimulatory polysaccharide isolated from the leaves of Diospyros kaki Thumb modulate macrophage via TLR2. Int J Biol Macromol 2015, 79, 971-982, doi:10.1016/j.ijbiomac.2015.06.023.

Reviewer 3 Report

Comments and Suggestions for Authors

The manuscript titled "Anti-metastatic effects of standardized polysaccharide fraction from Diospyros kaki leaves via GSK3β/β-catenin and JNK inactivation in human colon cancer cells" was interesting to me, the explanation and discussion of the results is clear. I consider it to be an interesting article for the readers of this Journal.

I only suggest that you update the references (from the last 5 years), since there are several very old ones that could be deleted (for example #12).

Author Response

REVIEWERS' COMMENTS:  

 Reviewer #3: The manuscript titled "Anti-metastatic effects of standardized polysaccharide fraction from Diospyros kaki leaves via GSK3β/β-catenin and JNK inactivation in human colon cancer cells" was interesting to me, the explanation and discussion of the results is clear. I consider it to be an interesting article for the readers of this Journal.

I only suggest that you update the references (from the last 5 years), since several very old ones could be deleted (for example #12).

We checked the manuscript and noticed that some sentences are detected similar or same with other published papers [The similarity rate is 40%]. You can see the results in the attachment (highlighted). We understand that some universal theories or methods might be used in many relative researches, but please try to rewrite those parts to make them literally different from other published papers during the minor revision.

Additionally, could you please provide the raw data for the figures included in the main text? This will allow us to accurately record the data for your manuscript.

Response: Thank you for your valuable advice. We deleted very old references and changed them to the most recent references within the past 5 years in the revised manuscript. In addition, we rewrote the content, did our best to reduce the similarity, and provided the raw data for the figures included in the main text.

Thank you very much for your advice.

Sincerely yours,

Kyung-Tae Lee, Ph.D.

Department of Pharmaceutical Biochemistry, College of Pharmacy

Kyung-Hee University

Dongdaemun-Ku, Hoegi-Dong 130-701, Seoul, Republic of Korea

Tel: +82-2-9610860, FAX: +82-2-9620860
